# Smoking-Mediated Upregulation of the Androgen Pathway Leads to Increased SARS-CoV-2 Susceptibility

**DOI:** 10.3390/ijms21103627

**Published:** 2020-05-21

**Authors:** Jaideep Chakladar, Neil Shende, Wei Tse Li, Mahadevan Rajasekaran, Eric Y. Chang, Weg M. Ongkeko

**Affiliations:** 1Department of Otolaryngology-Head and Neck Surgery, University of California San Diego, La Jolla, CA 92093, USA; jchaklad@ucsd.edu (J.C.); nshende@ucsd.edu (N.S.); wtl008@ucsd.edu (W.T.L.); 2Research Service, VA San Diego Healthcare System San Diego, La Jolla, CA 92161, USA; 3Department of Urology, VA San Diego Healthcare System, University of California San Diego, La Jolla, CA 92093, USA; mrajasekaran@health.ucsd.edu; 4Urology Service, VA San Diego Healthcare System, San Diego, CA 92161, USA; 5Department of Radiology, Radiology Service, VA San Diego Healthcare System San Diego, University of California San Diego, La Jolla, CA 92093, USA; Eric.Chang2@va.gov; 6Radiology Service, VA San Diego Healthcare System San Diego, La Jolla, CA 92161, USA

**Keywords:** COVID-19, SARS-CoV-2, ACE2, smoking, androgen, TMPRSS2

## Abstract

The COVID-19 pandemic is marked by a wide range of clinical disease courses, ranging from asymptomatic to deadly. There have been many studies seeking to explore the correlations between COVID-19 clinical outcomes and various clinical variables, including age, sex, race, underlying medical problems, and social habits. In particular, the relationship between smoking and COVID-19 outcome is controversial, with multiple conflicting reports in the current literature. In this study, we aim to analyze how smoking may affect the SARS-CoV-2 infection rate. We analyzed sequencing data from lung and oral epithelial samples obtained from The Cancer Genome Atlas (TCGA). We found that the receptor and transmembrane protease necessary for SARS-CoV-2 entry into host cells, ACE2 and TMPRSS2, respectively, were upregulated in smoking samples from both lung and oral epithelial tissue. We then explored the mechanistic hypothesis that smoking may upregulate ACE2 expression through the upregulation of the androgen pathway. ACE2 and TMPRSS2 upregulation were both correlated to androgen pathway enrichment and the specific upregulation of central pathway regulatory genes. These data provide a potential model for the increased susceptibility of smoking patients to COVID-19 and encourage further exploration into the androgen and tobacco upregulation of ACE2 to understand the potential clinical ramifications.

## 1. Introduction

COVID-19 is the respiratory illness caused by the SARS-CoV-2 virus that is currently affecting millions of people worldwide and putting extraordinary pressure on public health systems everywhere. Understanding the mechanism of this viral infection is important for developing preventative measures against infection and treatments for the disease. SARS-CoV-2 is a novel coronavirus which is part of the coronavirus family. It is related to the SARS-CoV and MERS-CoV viruses, which cause Severe Acute Respiratory Syndrome (SARS) and Middle East Respiratory Syndrome (MERS), respectively [1]. SARS-CoV-2 is enveloped, with a single-stranded positive sense RNA genome [2]. The viral envelope bears transmembrane spike proteins as well as other proteins.

In SARS-CoV-2, host recognition is carried out by the spike protein on the surface of the viral envelope. The spike protein binds to the ACE2 protein in human cells. After this, the spike protein can be cleaved by the serine protease TMPRSS2 [3,4]. Next, the fusion of the viral envelope with the membranes in the host cell allows for viral entry into the cell [5]. This process is vital for the entry of SARS-CoV-2 into human host cells, and thus it plays an integral role in COVID-19 infection and disease progression. 

COVID-19 also has a strong immunological component, where poor outcomes have recently been associated with cytokine storms and a hyperinflammatory immune system [6]. Case studies of COVID-19 patients show that previous and current smokers experience more severe infections than non-severe infections [7]. Additionally, current and former smokers more commonly require ventilation and are admitted to the ICU [8,9]. Since ACE2 is also an immunomodulator, its dysregulation could be essential for COVID-19 outcomes. ACE2 degrades angiotensin II into angiotensin (1–7) in the renin-angiotensin pathway, which, among other functions, is critical for the regulation of inflammation [10]. Angiotensin (1–7) has been shown to reduce inflammation in the lungs by binding to AT_2_ and Mas receptors [3]. Angiotensin II, meanwhile, activates the AT_1_ receptor, which promotes inflammation in the lungs [3]. Thus, the removal of angiotensin II seems to reduce inflammation [11]. However, this effect does not hold true in other parts of the body, as the activation of AT_1_ on immune cells prevents the polarization of macrophages and T-cells into pro-inflammatory types [12]. Furthermore, different and sometimes opposite effects from those in the lungs have been reported in the kidneys [13]. Since SARS-CoV-2 leads to the downregulation of ACE2, the resulting increase in angiotensin II could have significant effects on the immune system [14].

Cigarette smoke has been associated with an increased susceptibility to COVID-19 [15], although the mechanism for this is unclear. Cigarette smoke was found in some studies to increase ACE2 expression in the lungs of mammals [15,16,17]. Other analyses have found the co-expression of ACE2 and TMPRSS2 in normal human lung epithelial cell lines but did not analyze the effects of smoking on this interaction [18,19]. There have also been reports that smoking can lead to increases in androgen hormones, such as testosterone. The androgen receptor has been found to increase the expression of TMPRSS2 [17]. The modulation of TMPRSS2 by sex steroids has been highlighted as a possible mechanism that contributes to disparities in the SARS-CoV-2 infection rates between males and females [20]. However, the combination of ACE2 and TMPRSS2 expression in the context of smoking and their modulation by hormones has not yet been studied. This paper aims to investigate the expression of TMPRSS2, ACE2, and related genes in smokers and determine if the link between smoking and these genes could help explain smokers’ apparent increased vulnerability to COVID-19.

## 2. Results

### 2.1. Analysis in Lung Epithelium

RNA-sequencing data for lung epithelial solid tissue normal samples were obtained from The Cancer Genome Atlas (TCGA). Sequencing data were used to identify significant differences in gene expression between groups, namely smoking and nonsmoking patient samples. To investigate the possible effect of smoking in SARS-CoV-2, we compared the expression of ACE2 in current smokers versus past smokers (no data were available for never-smokers), as shown in Figure 1A. ACE2 was found to be significantly upregulated in current smokers. To investigate a possible functional relevance for the observed ACE2 upregulation, the Gene Set Enrichment Analysis (GSEA) was used to correlate ACE2 expression in current smokers to immune pathway dysregulation. ACE2 upregulation was significantly correlated to pathways related to immune cell population dysregulation, as shown in Figure 1B. 

Using the enrichment scores assigned to the significant pathways by GSEA, immune cell population sizes were inferred from gene expression data, and ratios of relevant cell populations were calculated, as shown in Figure 1C. ACE2 upregulation was correlated to higher dendritic cell, mast cell, and CD4+ T cell populations and lower macrophage, monocyte, CD8+ T cell, B cell, and neutrophil populations. These results indicate that smoking-related ACE2 upregulation may have a significant effect on immune response, which may be pivotal in relation to any SARS-CoV-2 response. 

Specifically, the impairment of antigen-processing and -presenting cells may decrease the recognition of the foreign pathogen. In order to further explore this relationship, differential expression was performed on the TMPRSS2 in current smoker vs. past smoker patients, as TMPRSS2 is critical for SARS-CoV-2 entry into the cell. TMPRSS2 was also upregulated in current smokers compared to past smokers, as shown in Figure 1D. The co-upregulation of ACE2 and TMPRSS2 in current smoker patients suggests a possible influence of smoking on patient susceptibility to SARS-CoV-2 infection via an increased likelihood of viral entry into lung epithelial cells.

In order to elucidate a mechanism for smoking-mediated susceptibility to SARS-CoV-2 infection, we utilized GSEA to compare the enrichment of pertinent pathways to ACE2 and TMPRSS2 expression. Of these relevant pathways, those involved in androgen signaling were emphasized. It was previously found that androgen signaling increases TMPRSS2 expression [17], and smoking leads to increases in androgen hormones [15]. Therefore, a potential mechanism for smoking-mediated susceptibility to SARS-CoV-2 may be the increased activity of the androgen signaling pathway paired with increased ACE2 expression. We discovered that ACE2 upregulation was also correlated to the positive enrichment of key androgen receptor pathways, as shown in Figure 2A. Of the core enriched genes in these pathways, many are the central regulators of said pathways, as they are shown to interact with a myriad of other core enriched genes, as shown in Figure 2B. TMPRSS2 expression is also correlated with the positive enrichment of androgen pathways, as shown in Figure 2C. However, fewer core enriched genes have high enrichment scores alongside TMPRSS2 upregulation, as shown in Figure 2D. Of these genes, HDAC6, CTNNB1, and SMARCA4 have relatively high enrichment scores and interact with many other core enriched genes. These results indicate that both ACE2 and TMPRSS2 upregulation in smoking patients are correlated to the overall upregulation of androgen pathways and to the upregulation of the central regulators of androgen pathways. This further lends support to our proposed potential mechanism of smoking-mediated susceptibility to SARS-CoV-2 infection in epithelial cells.

### 2.2. Analysis in Oral Epithelia

The primary entry route of the SARS-CoV-2 virus is through inhalation, with the virus eventually gaining access to the respiratory epithelium. However, there are several reports that this virus can also gain entry through other means, including through the oral route. For this reason, we investigated the expression of ACE2 and TMPRSS2 in oral epithelial cells. RNA-sequencing data for oral epithelial solid tissue normal samples were obtained from TCGA and were used to compare the gene expression between smoking and nonsmoking patients. ACE2 and TMPRSS2 were significantly upregulated in smokers versus nonsmokers, as shown in Figure 3A,B. ADAM17, a key mediator of ACE2 activity [21], and the androgen receptor gene were also upregulated in smokers, as shown in Figure 3C. These data suggest that the hypothesis we previously tested using lung epithelium samples may also extend to the oral epithelium. 

To further investigate the hypothesis in oral epithelial cells, GSEA was used to compare ACE2 and TMPRSS2 expression to androgen pathway activity in smoker oral epithelial cells. ACE2 expression was correlated to the enrichment of many core androgen pathway genes, including EP300, CDK6, SGK1, HDAC1, and NCOR1, as shown in Figure 4A. TMPRSS2 expression was correlated to the enrichment of ARRB2, CDK6, ID2, and EP300, as shown in Figure 4B. To investigate whether these core enriched genes are significantly dysregulated between smoking and nonsmoking samples, we used the Kruskal–Wallis test (*p* < 0.05). Of the core enriched genes with the most interactions with other core enriched genes in androgen pathways, ARRB2, CDK6, and ID2 were all upregulated in smokers. The knockdown of ARRB2 inhibits AR in androgen-dependent cells [22]. CDK6 has been shown to enhance AR activity by inducing its transcriptional activity in the presence of dihydrotestosterone [23]. Generally, androgen signaling has been shown to regulate the production of CDKs at the transcriptional and post-transcriptional levels [24]. The upregulation of CDK6 and CDK11 also suggests overall androgen pathway enrichment. Together, these results show that evidence of the smoking-mediated susceptibility to SARS-CoV-2 may also be present in oral epithelial cells, as ACE2 and TMPRSS2 upregulation in smoking patients is correlated to overall androgen pathway upregulation. 

## 3. Discussion

COVID-19 has been proven to be a dangerous and far-reaching disease as the number of infections and deaths continues to rise worldwide. Although the majority of patients eventually recover, the world has not seen this magnitude of devastation in such a short period of time in recent memory. The long-term effects of this virus are also unknown at this time, although it is believed that many patients will have significant sequelae from this infection. Therefore, aside from prevention from being infected, investigating the factors that determine susceptibility to COVID-19 infection and the mechanisms underlying these factors is vital to keeping SARS-Cov-2 in check. This study investigated one of these factors—smoking—and suggested a possible mechanism for smoking-mediated susceptibility to COVID infection.

We first analyzed lung epithelial tissue sequencing data, as this is the most common site of COVID infection. Our initial analysis revealed an upregulation of ACE2 alongside TMPRSS2 in smoking patients. Additionally, ACE2 expression in smoking patients was correlated to significant immune modulation. Together, these results suggest a possible role of smoking in SARS-CoV-2 infection, as the host cells show an upregulation of these two genes as well as decreased immune surveillance by antigen-presenting and -processing cells. It is already known that cigarette smoke alters androgen pathway signaling, and that androgens increase the expression of TMPRSS2. Taken together, we hypothesized that a smoking-mediated increase in androgen pathway signaling causes an increase in TMPRSS2 expression and activity alongside ACE2 upregulation, thereby increasing host cell susceptibility to SARS-CoV-2 entry. To investigate this hypothesis further, we analyzed the expression of androgen pathway genes in smoking patient samples. This analysis revealed that the enrichment of androgen pathways corresponded with the upregulation of ACE2 and TMPRSS2. 

To extend our model to a wider range of sites that may be the point of entry for SARS-Cov-2, we performed a similar analysis using oral epithelial tissue sequencing data. Similar to the lung epithelium, the oral epithelium exhibited an overexpression of ACE2 and TMPRSS2 in smoking patient samples. An analysis of androgen pathway expression in smoking patients showed an upregulation of key genes that are central regulators of the pathway. Although this effect is not as drastic as it is in the lung epithelium, our data show that our hypothesized mechanism for increased SARS-CoV-2 susceptibility may also be applied to the oral epithelial cells. In summary, our findings demonstrate a model for smoking-induced SARS-CoV-2 infection via the dysregulation of ACE2 and TMPRSS2 levels. This model may be useful in identifying at-risk patients and further our understanding as to why COVID-19 is such a dangerous and variable disease. Further investigation is necessary to investigate the interactions among tobacco smoke, androgens, SARS-Cov-2, ACE2, TMPRSS2, and their associated genes, including the validation of our computational data using in vitro and in vivo experiments. 

## 4. Materials and Methods 

### 4.1. RNA-Sequencing Data Acquisition

RNA-sequencing data were obtained for 49 adjacent normal tissue samples of lung squamous cell carcinoma and for 22 adjacent normal tissue samples of head and neck squamous cell carcinoma from the Cancer Genome Atlas (TCGA) (https://portal.gdc.cancer.gov/legacy-archive/search/f) on 5 August 2018. Clinical data, such as smoking history, for each patient were downloaded from the Broad Institute Firehose website (https://gdac.broadinstitute.org/). Genomic alteration data for each patient were obtained from the Broad Institute TCGA Genome Data Analysis Center’s (http://gdac.broadinstitute.org/runs/analyses__latest/reports/) analysis report (2016).

### 4.2. Differential Expression Analysis 

Differential expression analysis was performed to compare the gene expression between groups of interest in solid tissue normal samples. Kruskal–Wallis testing was then performed to find associations between the expression of the genes and the smoking status of patients. Patients who were smokers and nonsmokers were compared, as well as patients who were current smokers and past smokers. Associations were found to be significant if *p* < 0.05. 

### 4.3. Immune Pathway Association Using GSEA

Gene Set Enrichment Analysis (GSEA) was used to analyze all gene sets (pathways and signatures) for enrichment with respect to gene expression. Biological pathways and signatures were obtained from the Molecular Signature Database (MSigDB). The gene expression data for each of the five genes were inputted as continuous variables in the phenotype file. The gene expression dataset had expression values of all genes in counts per million (CPM). Pearson correlation was used to correlate gene expression with the expression of genes in a pathway to produce a ranked list.

## Figures and Tables

**Figure 1 ijms-21-03627-f001:**
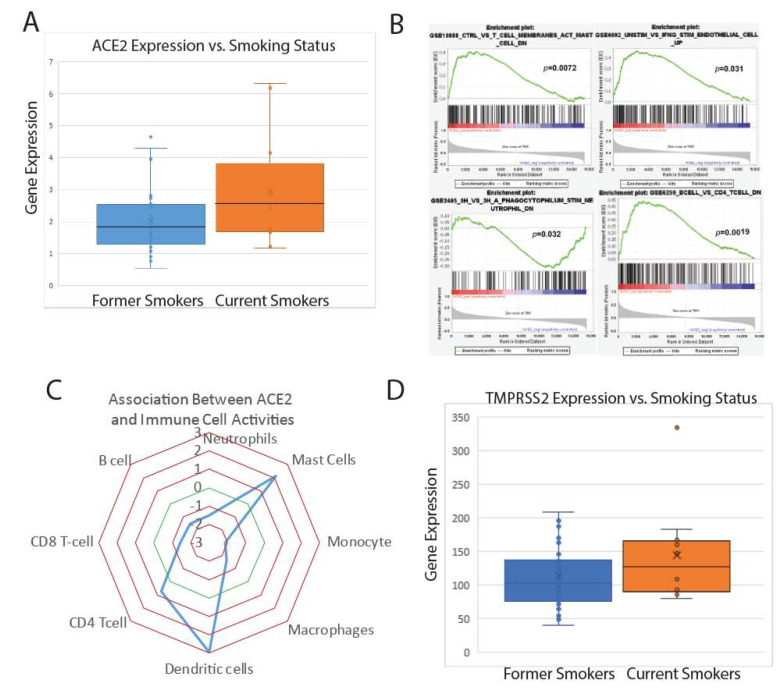
Initial analysis of gene expression and immune pathway dysregulation in lung tissue samples. (**A**) ACE2 expression in current vs. former smokers. (**B**) GSEA plots of pertinent immune pathway enrichment alongside ACE2 expression in current smokers. (**C**) Inferred immune cell population ratios based on GSEA data. (**D**) TMPRSS2 expression in current vs. former smokers.

**Figure 2 ijms-21-03627-f002:**
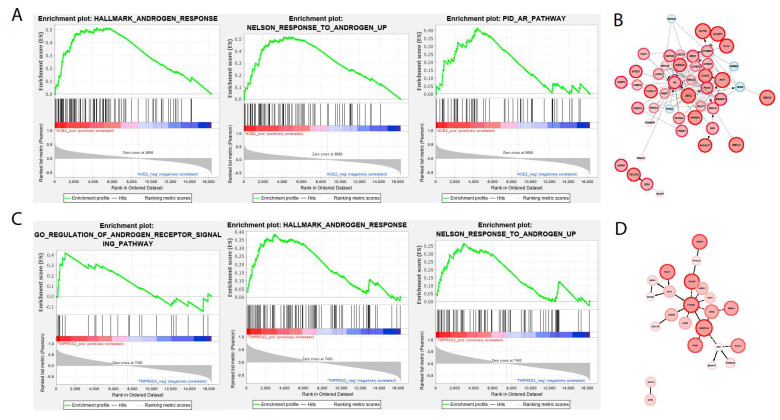
Relation between ACE2 and TMPRSS2 expression and androgen pathway activity. (**A**) GSEA of ACE2 expression compared to relevant androgen pathways. (**B**) Reactome Fi plot of core enriched genes based on the GSEA data in Figure 2A. (**C**) GSEA of TMPRSS2 expression compared to relevant androgen pathways. (**D**) Reactome Fi plot of core enriched genes based on GSEA data in Figure 2C. For Figure 2B,D, red nodes indicate positive enrichment scores, blue nodes represent negative enrichment scores, and node size indicates enrichment score magnitude.

**Figure 3 ijms-21-03627-f003:**
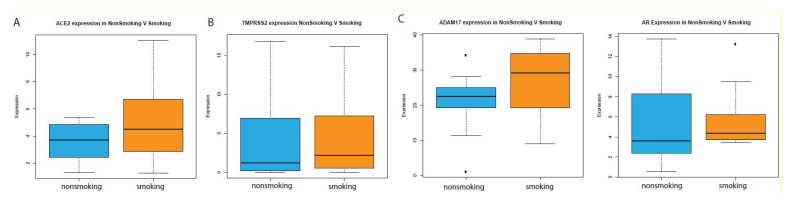
Analysis of gene expression in oral tissue samples. Pertinent dysregulated genes include the coreceptors (**A**) ACE2 and (**B**) TMPRSS2 as well as (**C**) genes indicative of androgen pathway activity.

**Figure 4 ijms-21-03627-f004:**
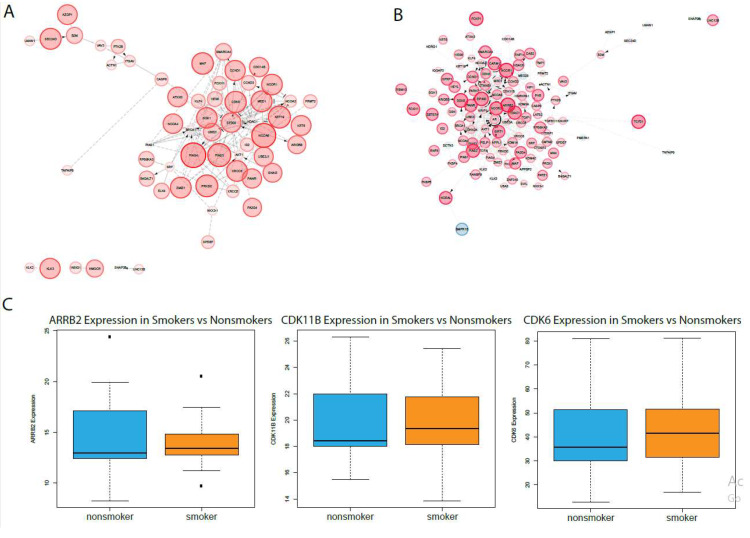
Analysis of androgen pathway expression in oral smoker samples. Core enriched genes from GSEA comparing (**A**) ACE2 and (**B**) TMPRSS2 expression to androgen pathway activity were plotted using Reactome Fi. Red nodes indicate positive enrichment scores, blue nodes indicate negative enrichment scores, and node size indicates enrichment score magnitude. (**C**) Kruskal–Wallis plots of expression of core enriched genes from the two sets of GSEAs.

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
