# Peer review of "Smoking-Mediated Upregulation of the Androgen Pathway Leads to Increased SARS-CoV-2 Susceptibility"

_ijms, 2020, doi:10.3390/ijms21103627_

Round 1

Reviewer 1 Report

In this paper, Chakladar et al investigated the association of ACE2 and TMPRSS2 up-regulation, the host proteins required for SARS-CoV2 with smoking mediated susceptibility of COVID infection. Recently, there is a great surge in studying association of various risk factors including smoking with increased susceptibility of SARS-COV2 infection. The studies are well performed and the manuscript is well written, with adequate figures added. I have only a few minor comments. Overall, I recommend this article for publication. I would expect authors to address the outstanding minor issues as highlighted below in the comments before publication.

Line 57 and 58. Please provide references here.

General comment: Is it feasible for authors to assess and compare the expression of ACE2 and TPMRSS2 between smokers vs non-smokers in SARS-CoV2 patients.

Author Response

 In this paper, Chakladar et al investigated the association of ACE2 and TMPRSS2 up-regulation, the host proteins required for SARS-CoV2 with smoking mediated susceptibility of COVID infection. Recently, there is a great surge in studying association of various risk factors including smoking with increased susceptibility of SARS-COV2 infection. The studies are well performed and the manuscript is well written, with adequate figures added. I have only a few minor comments. Overall, I recommend this article for publication. I would expect authors to address the outstanding minor issues as highlighted below in the comments before publication.

We thank the reviewers for their comments and their constructive critiques of our submission. 

Line 57 and 58. Please provide references here.

We have added the appropriate references for the information cited in lines 57 and 58.

General comment: Is it feasible for authors to assess and compare the expression of ACE2 and TPMRSS2 between smokers vs non-smokers in SARS-CoV2 patients.

Although this is a very good suggestion and something we would really like to do, unfortunately, it is not possible for us to study sequencing data from SARS-CoV-2 patients as there are no publicly available sequencing data for such patients at this time.  We intend to analyze sequencing data from COVID-19 patients once these are available for a follow up publication.

Reviewer 2 Report

In this manuscript the authors investigate the possible mechanisms by which cigarette smoking increases the susceptibility to the COVID-19 infection, even though the relationship between smoking and COVID-19 outcome is still controversial.

The authors then analyze the expression of ACE2 and TMPRSS2, that represent the entry keys of the virus, using the data from The Cancer Genome Atlas (TCGA) in oral and lung epthelial samples of smokers and non-smokers.

They find an upregulation of ACE2 and TMPRSS2 in smokers that correlates with an increased expression of genes involved in the androgen pathway.

Therefore,  these findings would account for the poorer outcome of COVID-19 cigarettesmokers infected patients and also of males with respect to females.

The results are clear and interesting, and appropriate to the actul linterest of the scientific community

Reviewer 3 Report

The paper is very interesting as the authors try to prove that upregulation of the androgen 2 pathway may increase the susceptibility to covid-19.

1. There are some missing discussions of the findings of world statistical studies about the number of smokers who were infected by covid-19 compared to the number of non-smokers. You may refer also to:

Vardavas, C. I., & Nikitara, K. (2020). COVID-19 and smoking: A systematic review of the evidence. Tobacco induced diseases, 18.

  2. The authors should also highlight more their contribution compared to results reported in:   Lukassen, S., Chua, R.L., Trefzer, T., Kahn, N.C., Schneider, M.A., Muley, T., Winter, H., Meister, M., Veith, C., Boots, A.W. and Hennig, B.P., 2020. SARS‐CoV‐2 receptor ACE2 and TMPRSS2 are primarily expressed in bronchial transient secretory cells. The EMBO journal.   Cai, G., Bossé, Y., Xiao, F., Kheradmand, F., & Amos, C. I. (2020). Tobacco smoking increases the lung gene expression of ACE2, the receptor of SARS-CoV-2. American journal of respiratory and critical care medicine, (ja).   Stopsack, K. H., Mucci, L. A., Antonarakis, E. S., Nelson, P. S., & Kantoff, P. W. (2020). TMPRSS2 and COVID-19: Serendipity or Opportunity for Intervention?.   3. I think section 4 should be in place of section 2.

Author Response

The paper is very interesting as the authors try to prove that upregulation of the androgen 2 pathway may increase the susceptibility to covid-19.

  1. There are some missing discussions of the findings of world statistical studies about the number of smokers who were infected by covid-19 compared to the number of non-smokers. You may refer also to: Vardavas, C. I., & Nikitara, K. (2020). COVID-19 and smoking: A systematic review of the evidence. Tobacco induced diseases, 18.
  2. The authors should also highlight more their contribution compared to results reported in: Lukassen, S., Chua, R.L., Trefzer, T., Kahn, N.C., Schneider, M.A., Muley, T., Winter, H., Meister, M., Veith, C., Boots, A.W. and Hennig, B.P., 2020. SARS‐CoV‐2 receptor ACE2 and TMPRSS2 are primarily expressed in bronchial transient secretory cells. The EMBO journal. Cai, G., Bossé, Y., Xiao, F., Kheradmand, F., & Amos, C. I. (2020). Tobacco smoking increases the lung gene expression of ACE2, the receptor of SARS-CoV-2. American journal of respiratory and critical care medicine, (ja).   Stopsack, K. H., Mucci, L. A., Antonarakis, E. S., Nelson, P. S., & Kantoff, P. W. (2020). TMPRSS2 and COVID-19: Serendipity or Opportunity for Intervention?. 

Thank you very much for pointing out these relevant papers that we are happy to include in our manuscript.

To address item 1, we have added relevant clinical statistics of current and former smokers from studies referenced in the suggested review. These additions are in lines 48-51 of the revised manuscript.

To address item 2, we have added comparisons to the results reported in the suggested papers in lines 64-66 (Lukassen et al and Cai et al) and lines 68-69 (Stopsack et al).

Round 2

Reviewer 3 Report

With these new details and literature, the paper is now clearer.

I checked now the methods, they are good.

I suggest the paper to be published in IJMS.